# Optical Detection of Denatured Ferritin Protein via Plasmonic Gold Nanoparticles Exposure through Aminosilane Solution

**DOI:** 10.3390/nano9101417

**Published:** 2019-10-04

**Authors:** Monique J. Farrell, Robert J. Reaume, Erin A. Jenrette, Jasmine Flowers, Kevin C. Santiago, Kyo D. Song, Aswini K. Pradhan

**Affiliations:** 1Center for Materials Research, Norfolk State University, Norfolk, VA 23504, USA; mfarrell@nsu.edu (M.J.F.); rreaume@nsu.edu (R.J.R.); ejenrette@nsu.edu (E.A.J.); jflowers@nsu.edu (J.F.); ksantiago@nsu.edu (K.C.S.); ksong@nsu.edu (K.D.S.); 2Advance Material Solution, 2608 Horse Pasture Road, Suite 101, Virginia Beach, VA 23453, USA

**Keywords:** biosensor, gold nanoparticles, denatured protein, visual detection, ferritin, degraded protein

## Abstract

The presence of denatured proteins within a therapeutic drug product can create a series of serious adverse effects, such as mild irritation, immunogenicity, anaphylaxis, or instant death to a patient. The detection of protein degradation is complicated and expensive due to current methods associated with expensive instrumentation, reagents, and processing time. We have demonstrated here a platform for visual biosensing of denatured proteins that is fast, low cost, sensitive, and user friendly by exploiting the plasmonic properties of noble metal nanoparticles. In this study we have exposed artificially heat stressed ferritin and gold nanoparticles to 3-aminopropyl triethoxysilane, which degrades the protein by showing a systematic blue shift in the absorbance spectra of the gold nanoparticle/ferritin and aminosilane solution. This blue shift in absorbance produces a detectable visual color transition from a blue color to a purple hue. By studying the Raman spectroscopy of the gold nanoparticle/ferritin and aminosilane solution, the extent of ferritin degradation was quantified. The degradation of ferritin was again confirmed using dynamic light scattering and was attributed to the aggregation of the ferritin due to accelerated heat stress. We have successfully demonstrated a proof of concept for visually detecting ferritin from horse spleen that has experienced various levels of degradation, including due to heat stress.

## 1. Introduction

The area of pharmaceutics is a multibillion-dollar industry with drug products ranging from proteins and antibodies to hormones. Detection of these biomolecules in their denatured state is extremely important as the denatured or aggregated structures can cause mild irritation, immunogenicity, anaphylaxis, or instant death [1,2]. Currently, stability of these biological molecules can be determined via techniques such as size exclusion chromatography, dynamic light scattering, and SDS-PAGE gels [3,4,5]. However, the majority of these methods are costly and not easily mobilized, making them impractical. Therefore, it is worthwhile to design a simple and effective method for the visual detection of these denatured biomolecules. This work proposes a unique and inexpensive method for the visual detection of denatured proteins using the plasmonic behavior of gold nanoparticles.

Due to ease in processing of gold nanoparticles, their high biocompatibility, and diverse optical properties, these nanoparticles are the working material of choice for this study [6,7,8]. In addition, as a result of highly active and sensitive local surface plasmon properties, visual/optical applications in biological and environmental sensing are extremely feasible [9]. Surface plasmons are the electronic oscillations at the interface between a metal and dielectric which can be excited by incident light waves [10]. For the purpose of this study we utilize localized surface plasmons which occur in metallic structures that are confined in at least one dimension in the nanoscale. Specifically, the electronic cloud and plasmonic oscillations of the gold nanoparticles (AuNps) used in this work are central to the particle itself, as opposed to the metallic/dielectric interface [11,12]. This centralized oscillation is therefore extremely sensitive to the surface of the nanoparticle and is a key factor that allows for the detection of analytes bound to or near the surface of the AuNps.

Based on reports with applications in the biomedical field, the aggregation of gold nanoparticles can be exploited to detect biomolecules through fluctuations in the localized surface plasmon frequencies [13,14,15,16,17]. Within this study, the color of the solutions is altered via induction of gold nanoparticle aggregation due to exposure of the AuNps to 3-aminopropyl triethoxysilane (APTES). By manipulating inter-particle spacing and surface chemistry, our work fine tunes the absorbance spectra of the gold nanoparticles for applications in visual detection [18]. These shifts in the optical absorbance correspond to visual color changes ranging from red to purple and blue. The interactions and changes in the localized surface plasmon frequency of the AuNps due to changes in the surrounding solution are the primary governing physics that is used in this study.

For the purpose of this work, ferritin type-I from horse spleen was used as the target analyte. Ferritin is a ubiquitous protein responsible for the regulation of iron within an organism [19]. Iron is a critical metal to life as it facilitates growth and metabolic pathways within cells, however excess iron is detrimental making ferritin crucial for life processes [20]. Ultimately, we have utilized the size and surface sensitivity of gold nanoparticles for the detection of denatured ferritin as the analyte. The development of a highly precise and accurate in-solution biosensor that produces a visual color change detectable by the human eye will provide evidence and an additional means to ensure product quality. This research will ultimately increase patient’s quality of life by allowing care providers to instantly determine the stability of pharmaceutical drug products before administration to patients.

## 2. Experimental Section

Gold chloride trihydrate (HAuCl_4_∙3H_2_O, metal basis) was obtained from Sigma-Aldrich (Milwaukee, WI, USA) with a 99.9% purity. A 99% ACS purity trisodium citrate reagent was purchased from Alfa Aesar (Ward Hill, MA, USA). APTES was procured from Sigma-Aldrich with composition purity greater than 98%. Ferritin, type-I from horse spleen, was purchased from Sigma-Aldrich and stored in a 0.15 M sodium chloride (NaCl) sterile filtered aqueous solution. Without employing any further purification methods, chemicals were dissolved in deionized water (DI H_2_O).

We used a simple one pot synthesis method to synthesize the gold nanoparticles as described by Grabar et al. [21]. The method follows as: First, gold chloride and trisodium citrate were dissolved in separate vials containing DI H_2_O, followed by dissolving 115 mg of trisodium citrate in 10 mL of DI H_2_O and 40 mg of the gold chloride contained in a separate vial. Then the gold (Au) precursor solution was heated to approximately 100 °C, and trisodium citrate was quickly injected into the reaction vial. The solution was low-boiled for 10 min, and then the gold nanoparticle solution was placed in an ice bath for 5 min under constant mixing. In order to increase the working sample volume, the gold nanoparticle solution was diluted 1:2 in DI H_2_O.

The ferritin from horse spleen was diluted in deionized water for all concentrations used in this study. A ferritin concentration of 1.14 × 10^−7^ M was utilized for the accelerated heat stress component of this study. To denature the samples via an artificial heat stress, the ferritin solution was placed in a hot water bath set to 70 °C. It is important to note that the temperature of the water bath was recorded as opposed to the protein solution. The aminosilane solutions were prepared based on a volume-to-volume ratio and diluted using deionized water. From the denatured ferritin solution, 100 μL were mixed with 0.8 mL of the gold nanoparticle (AuNp) solution. To the AuNp and ferritin complex, 100 μL of various concentrations of APTES was then added. After mixing, the solutions were immediately characterized as described within the characterization section.

The LAMBDA 950 UV/Vis/NIR Spectrophotometer (PerkinElmer, Shelton, CT, USA) was utilized to obtain the absorbance spectra for this work. The peak wavelength of the absorbance spectra was used to correlate the hues of the gold nanoparticle/protein and APTES solutions. A 2 nm slit, with scan speed of 480 nm/min and a range of 400–800 nm, was employed. To determine the size and size distribution of the ferritin pre and post heat stress, a dynamic light scattering technique was used via the Nanotrac Wave by Microtrac (Montgomeryville, PA, USA) which was operated with run times lasting 60 s, coupled with a 780 nm laser diode. To further characterize the ferritin, the EZRaman-I series high sensitivity portable Raman analyzer from Enwave Optronics (Irvine, CA, USA) as used. To determine secondary and quaternary structural changes within the denatured ferritin, circular dichroism spectra were obtained using the JASCO J-815 CD Spectrophotometer (Easton, MD, USA). The sample runs were recorded at 20 °C and 70 °C utilizing a 200 nm/min scan speed. Transmission electron microscopy (TEM) images were taken of the gold nanoparticle and APTES samples using the JEOL JEM-2100F high resolution TEM (Peabody, MA, USA).

## 3. Results and Discussion

We have developed a unified method for the visual detection of denatured forms of proteins utilizing concentration-dependent aggregation of AuNps via APTES. From previously published work, it is known that the addition of an aminosilane to gold nanoparticles will induce extensive shifts in the absorbance spectra and can be exploited to visually detect stable proteins [22,23]. With the target goal of ferritin visual detection, a highly sensitivity experiment was employed to confirm viability of ferritin for the proposed study. Several concentrations of ferritin ranging from 1 × 10^−3^ M to 1 × 10^−6^ M were prepared and combined with the AuNp solution described in the experimental section. Each trial concentration was then exposed to 100 μL of APTES and the changes in the absorbance spectra as a function of the protein concentration were monitored. In Figure 1 we observe a systematic red shift as the concentration of the ferritin is decreased. We found a strong dependence of the peak wavelength of the nanoparticle solutions on the ferritin concentration displayed in Figure 1. With a decrease in the ferritin concentration, the gold nanoparticles were able to aggregate more extensively upon the addition of APTES, causing the systematic shift. From this experiment we were able to demonstrate that the aggregation of the AuNps when exposed to APTES is dependent on the concentration of ferritin in solution. Figure 1 confirms the sensitivity of this system to the presence of ferritin in solution and suggests the viability of visual detection of denatured ferritin using a similar mechanism. The inset in Figure 1 shows the sensitivity of the peak absorbance to APTES vs. ferritin concentration, which confirms that the aggregation of the AuNps is dependent on the concentration of ferritin in solution when exposed to APTES.

With the target goal of capturing protein denaturation visually, a system to artificially expose the ferritin to a heat stressed environment was created. To artificially heat stress the proteins, a water bath was preheated to 70 °C and monitored to maintain the set point temperature. This method of accelerated heat stress is common and has been used in several protein-based studies to accelerate a protein degradation [24,25,26,27]. Circular dichroism (CD) was employed to monitor the changes in the secondary and quaternary structures of ferritin as the sample was heat stressed. The control, or non-denatured trials, was measured at 20 °C. In Figure 2a, immediately upon exposure to the 70 °C temperature, we observe that that quaternary structure of the ferritin had broken down. This was indicated via a distinct loss of signal within the near-UV CD spectra as compared to the 20 °C trial. Taking a closer look at the subunits of the ferritin, consisting of alpha helixes, the CD spectra were also monitored in far-UV (190–250 nm) region as a temporal function of accelerated heat stress. It is noted that CD spectroscopy can determine secondary structure in the “far-UV” spectral region (190–250 nm).

In Figure 2b, the trial at 20 °C displays a spectra characteristic of a protein containing a majority of alpha helixes. In Figure 2b, we observe the characteristic double minima and singular maximum of the alpha helix secondary structure. As the ferritin is heat stressed, an increase in both minima and decrease in the maxima are observed, indicating the unfolding of the secondary structures within the protein samples. Figure 3 displays a simplified model of the denaturation pathway for ferritin as the 70 °C heat stress is applied. The native ferritin is a hollow spherical structure composed of 24 subunits in a quaternary structure [28]. As the heat stress is applied, the tertiary structure is lost and the secondary structure is preserved, but over time unfolds into the primary structure. 

To further characterize the denatured ferritin, the prepared ferritin solution was then exposed to the artificial heat stress and aliquots at 0, 1, 3, 6, 9, and 30 min were removed and mixed with 0.8 mL of an AuNp solution. Each time aliquot consisting of denatured ferritin was added to the gold nanoparticle solution and then measured using Raman spectroscopy. The addition of the gold nanoparticles to the denatured ferritin solution was performed to enhance the signal sent to the detector from low intensity trials due to the denaturing process. This experimental procedure was based on the concept of surface enhanced Raman spectroscopy (SERS) in which the signal of an analyte can be increased due to hot spots that form, most often between plasmonic particles and structures [29,30]. Figure 4a shows the resulting characteristic Raman spectra of the ferritin from horse spleen. Three distinct peaks were observed (293 cm^−1^, 391 cm^−1^, and 736 cm^−1^) and are comparable to previously reported Raman spectra corresponding to ferritin [31]. In order to quantify the extent of degradation, the intensity of the 736 cm^−1^ peak was plotted as a function of the accelerated heat stress exposure time as seen in Figure 4b. We observe a negative relationship between the intensity and the exposure time (see Figure 4b). This decrease in intensity is due to the protein denaturation and subsequent aggregation of the ferritin as the exposure time was increased, resulting in a lowered spectral signal. The Raman spectra in Figure 4a confirm the degradation of the protein via the loss of intensity with respect to the 736 cm^−1^ peak. The manner in which the degradation of the protein occurred was confirmed using a dynamic light scattering technique. Within this study, a denatured and non-denatured samples of the ferritin were characterized. In Figure 4c there is an increase in the size and size distribution of the denatured sample as opposed to the non-denatured sample. In this Figure we observe an initial protein size of about 10 nm, which increases to micron scale aggregates as the heat stress is applied. This type of denaturation observed in the ferritin protein has also been reported due to heat stress and other external stressors [32,33].

In order to evaluate the dependence of gold nanoparticle optical absorbance on different concentrations of APTES in the presence of non-denatured and denatured ferritin, the following study was executed. The ferritin solutions were denatured in a hot water bath at 70 °C and then added to 0.8 mL of an AuNp solution and exposed to 5%, 1%, 0.5%, and 0.2% APTES, respectively. The resulting solutions were then characterized via ultraviolet visible spectroscopy to determine the absorbance spectra and peak wavelength. Figure 5 displays the resulting spectra for the four different concentrations of APTES utilized, maintaining the same extent and range of ferritin degradation. In Figure 5a we observed a stable peak wavelength at 5.0% APTES over a 15 min exposure time. The concentration of APTES was then decreased five-fold producing a 1.0% APTES concentration and the experiment was repeated. Within Figure 5b we observed a systematic decrease in the peak absorbance wavelength as the heat stress exposure time was increased. In subsequent trials the concentration of APTES was decreased to 0.5% and 0.2%. In Figure 5c,d, the absorbance spectra display larger blue shifts as the ferritin is denatured. All trials containing less than 5.0% APTES experienced a similar trend with respect to a blue shift in the optical absorbance. However, it is important to note that although the trends observed were similar, the spectra varied with respect to the peak wavelength rate of change. These differences facilitate the potential for visual detection of different concentrations of denatured proteins by varying the concentration of APTES.

By manipulating the absorbance spectra via variations within the APTES concentration, we directly target and induce color detection at precise concentrations of protein denaturation. Figure 6 displays images of the AuNps, ferritin, and APTES solutions and was used to capture the resulting hues of each trial. In Figure 6, all samples were exposed to the same artificial heat stress; however, the concentrations of APTES employed were different. In Figure 5a the negligible changes observed within the absorbance spectra of the ferritin and AuNps solution exposed to 5.0% APTES corresponded to a consistent blue hue throughout all of the samples seen in Figure 6a. Inversely, as we decreased the APTES concentration we observe progression from a blue hue to a purple hue as the ferritin is denatured. At 1.0% APTES, the color change or detection indicator did not occur until 9 min of ferritin exposure to the external heat source displayed in Figure 6b. This visual color changes from blue to purple at 6 min and 3 min for the trials consisting of 0.5% and 0.2% APTES, respectively. Figure 6 strongly suggests the potential to tailor this visual detection system to produce a distinct color change at varying degrees of degradation by varying the concentrations of APTES in solution.

To further the objective of creating a unified visual detection method for denatured proteins, a systematic study was developed. The objective was to understand how different concentrations of APTES would affect a stable gold nanoparticle and ferritin complex. Figure 7a,c displays the absorbance spectra of gold nanoparticles and ferritin exposed to varying concentration of APTES ranging from 0.1% to 10%. We observed a systematic red shift and then a strong blue shift as the concentration of APTES is increased beyond 0.2%. This red shift and subsequent blue shift is clearly visible within Figure 7b where we see a 100 nm increase in the wavelength from 0.1 to 0.2%. This trend becomes negative due to a decrease of 30 nm in wavelength as the concentration of APTES is increased from 0.2% to 10%. This information is critical when designing the visual detection parameters, as an optimal concentration of APTES must be determined to induce the color change at specific degrees of degradation. For example, if a protein degrades very easily, or the threshold for degradation is small, then an optimal concentration would fall in the range of the steepest slope. For the instance of ferritin, APTES concentrations between 0.1% and 0.2% should be evaluated first for optimization. In contrast, if the protein has a high limit of degradation or a higher protein denaturation limit, then a higher concentration of APTES should be explored.

### Proposed Mechanism

The gold nanoparticle solution is relatively mono disperse and maintains an optical absorbance of about 500 nm at a size of 10–15 nm, which is comparable to literature for the corresponding wavelengths [34,35]. In Figure 8a,b, we observe monodisperse TEM samples of the gold nanoparticles with no APTES present. However, upon the addition of 0.1% APTES to the gold nanoparticle solution we detect higher order aggregate formation at the 200 nm scale (Figure 8c). In Figure 8d we see examples of dimers and higher order aggregates forming within the AuNps. These dimers are induced as a result of dipoles that form due to the presence of APTES, which causes the AuNps to aggregate [36]. APTES has the amine group that is electrostatically attracted to the negatively charged surface of the gold nanoparticles [37,38]. This resulting coulombic attraction due to the dipole in the gold nanoparticles increases the overall energy state of the particle. The gold nanoparticles aggregate stimulates a color change in the solution in order to decrease this energy state.

Based on the ferritin aggregation observed in Figure 4c, detailing the size and size distribution of denatured vs. non-denatured ferritin samples, the following mechanism for the detection is proposed. Without ferritin present in solution, the addition of 100 μL of APTES ranging from 0.2–5.0% will induce a visual color change from a red color to a blue hue. With addition of 100 μL of a 1.14 × 10^−7^ M ferritin concentration, exposure to this range of APTES concentrations will also produce a blue color. We argue that as the protein begins to denature, the ferritin agglomerates forming barriers that prevent the aggregation of the gold nanoparticles as a function of the extent of ferritin degradation as shown schematically in Figure 9. We propose that as the ferritin is denatured, it blocks the gold nanoparticles more effectively by closing the inter-particulate space and degree of aggregation. This causes a systemic blue shift on the application of heat to the protein. Similar phenomenon was observed for some other proteins, such as polynucleotides [18], bovine serum albumin [23], and glutathione peptides [39] with a reproduceable results. The current results are significantly important compared to other reports as we present the size and size distribution of denatured vs. non-denatured ferritin protein, which is a real constituent of our blood. It is anticipated that the proof-of-concept presented here is valid for other protein molecules as well. As the presence of aggregates is a reoccurring concern in the pharmaceutical industries these works propose an initial method for the visual detection of heat stressed proteins. Further studies are necessary in order to compare with other biological analytes, such as antibodies and antibody conjugates, using this system is necessary in order to understand if the visual detection is influenced by the other components of this therapeutic drug product.

## 4. Conclusions

We have demonstrated how the molecular interactions with the surface plasmons of gold nanoparticles can be exploited to visually detect denatured proteins in solution. We have provided the basis for a thorough investigation into the visual biosensing of denatured drug products utilizing concentration dependent induced aggregation of gold nanoparticles via 3-aminopropyl triethoxysilane. With the objective of removing the need for many of the costs associated with medical diagnostics, an effective visual method for the detection of denatured ferritin was created. Significantly systematic blue shifts in the absorbance spectra of gold nanoparticles associated with denatured protein in the presence of the aminosilane were observed corresponding to remarkable visual color changes in the solution. The product of this research will provide clinicians with an instant visual detection system to ensure stability of drug products before administration to patients, making this a high impact system for prevention of drug related casualties. The other components that include the addition of chemicals, lyophilized materials, the effect of mechanical and heat stresses encountered during manufacturing, freeze/thaw cycles, and prolonged storage influence the therapeutic drug product. It is very important to understand the pathway of aggregate formation in order to put in place processes that will help to minimize the process of protein aggregation at an early stage. The proof-of-concept presented in this work has tremendous scientific as well as medical values in order to understand if the visual detection is influenced even by the other components of this therapeutic drug product as described above. 

## Figures and Tables

**Figure 1 nanomaterials-09-01417-f001:**
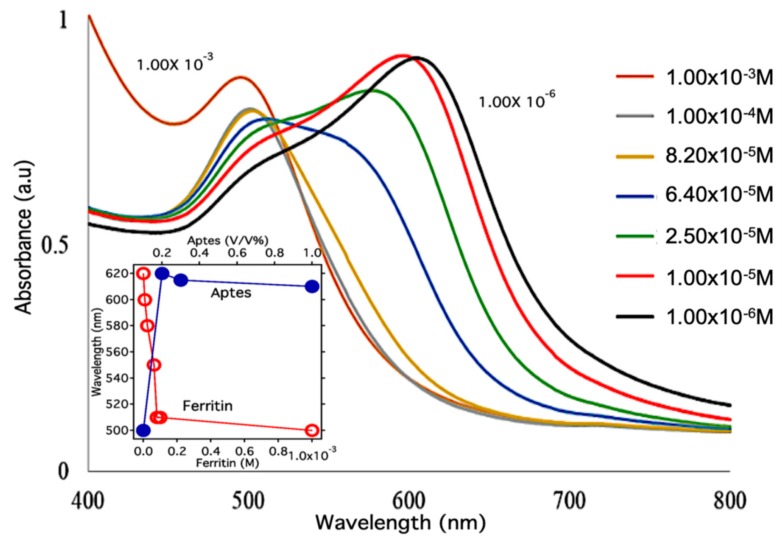
Ultraviolet visible absorbance spectra of gold nanoparticle colloidal solutions with various concentrations of ferritin ranging from 1 × 10^−3^ M to 1 × 10^−6^ M exposed to 3-aminopropyl triethoxysilane (APTES). The inset shows the sensitivity of the peak absorbance to APTES vs. ferritin concentration.

**Figure 2 nanomaterials-09-01417-f002:**
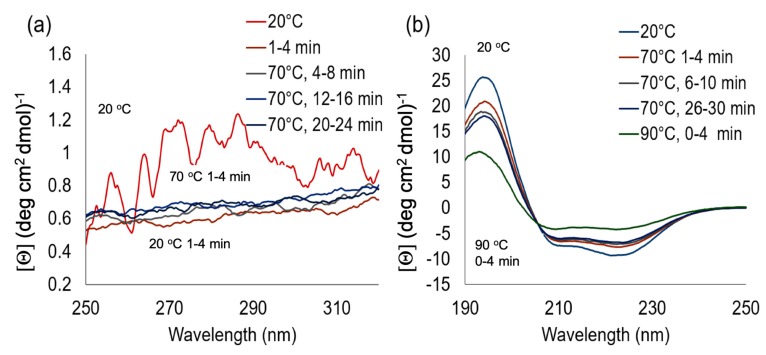
Displays the circular dichroism spectra for the ferritin protein at 20 °C and a 70 °C accelerated heat stress (**a**) 250–320 nm (near-UV) and (**b**) 190–250 nm (far-UV) spectral region.

**Figure 3 nanomaterials-09-01417-f003:**
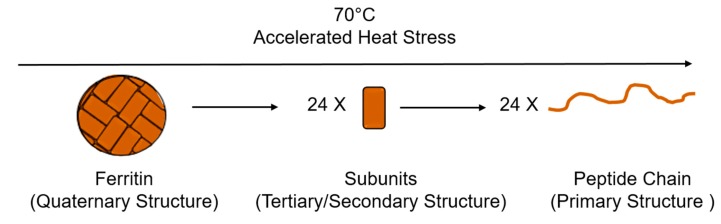
Displays the changes in the ferritin structure during the accelerated heat stress study, progressing from the quaternary structure, to secondary and ultimately primary structure.

**Figure 4 nanomaterials-09-01417-f004:**
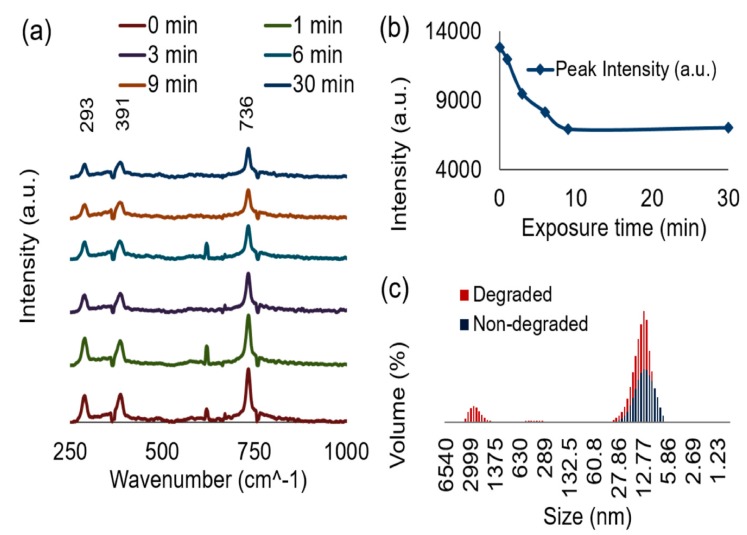
(**a**) Display of the Raman spectra of ferritin over a 30 min accelerated heat stress exposure time. (**b**) Shows the change in the intensity of the 736 cm^−1^ peak taken from the Raman spectra as a function of the exposure time. (**c**) Showcases the size and size distribution of the denatured and non-denatured ferritin samples.

**Figure 5 nanomaterials-09-01417-f005:**
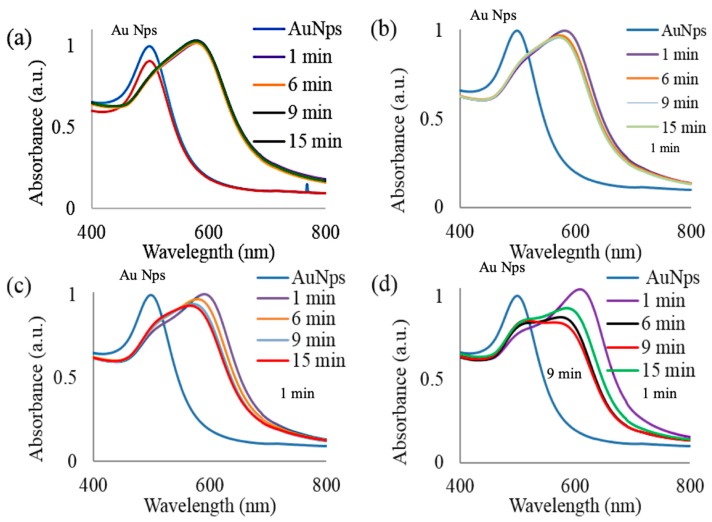
Displays the ultraviolet visible absorbance spectra of AuNps and APTES systems in the presence of heat stressed ferritin ranging from 1 to 15 min exposure times (**a**) 5% APTES, (**b**) 1% APTES, (**c**) 0.5% APTES, and (**d**) 0.2% APTES for denatured and non-denatured ferritin samples.

**Figure 6 nanomaterials-09-01417-f006:**
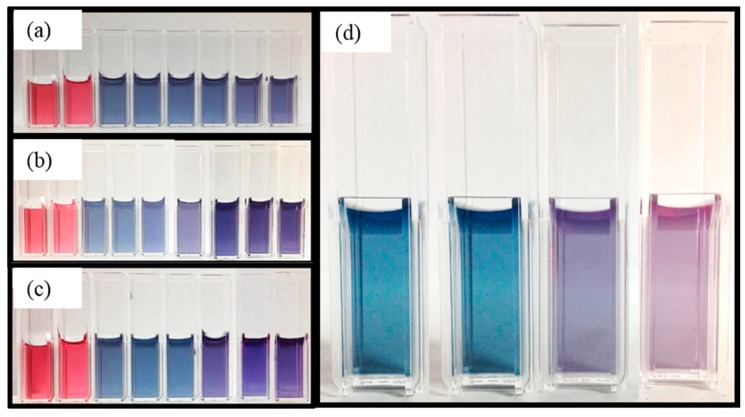
Displays resulting hues of trial solutions composed of AuNps, ferritin, and APTES. (**a**) 100 μL of 5.0% APTES. From left to right: AuNps, AuNps/ferritin, 0 min, 1 min, 3 min, 6 min, 9 min, 12 min, and 15 min (**b**) 100 μL of 1% APTES. From left to right: AuNps, AuNps/ferritin, 0 min, 1 min, 3 min, 6 min, 9 min detection, 12 min, and 15 min (**c**) 100 μL of 0.5% APTES. From left to right: AuNps, AuNps/ferritin, 0 min, 1 min, 3 min, 6 min detection, 9 min, and 15 min. (**d**) 80 μL of 0.2% APTES. From left to right: 0 min, 1 min, 3 min detection, and 6 min.

**Figure 7 nanomaterials-09-01417-f007:**
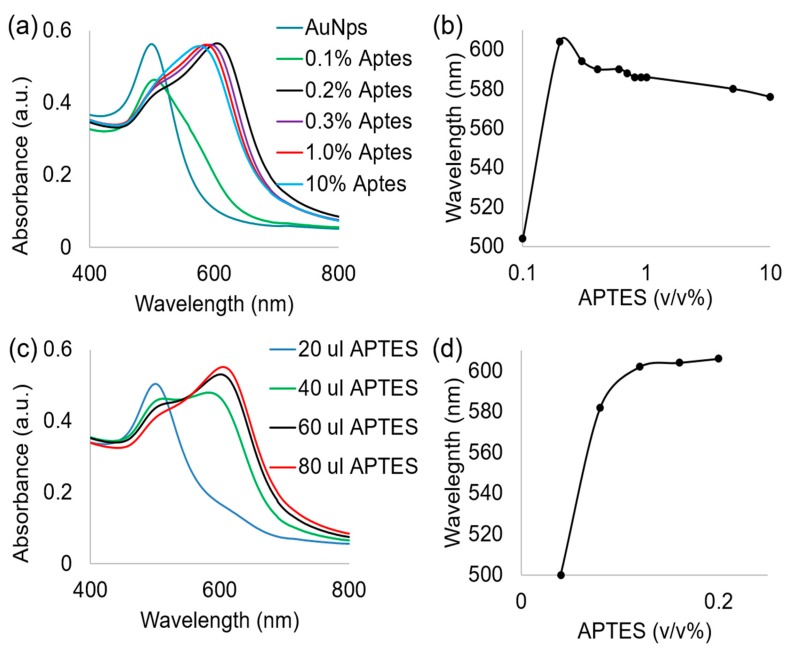
Displays the ultraviolet visible absorbance spectra of ferritin and AuNp system exposed to APTES and the absorbance peak wavelength as a function of the APTES concentration. (**a**,**b**) 0.1–10% APTES and (**c**,**d**) 0.04–0.2% APTES for denatured and non-denatured ferritin samples.

**Figure 8 nanomaterials-09-01417-f008:**
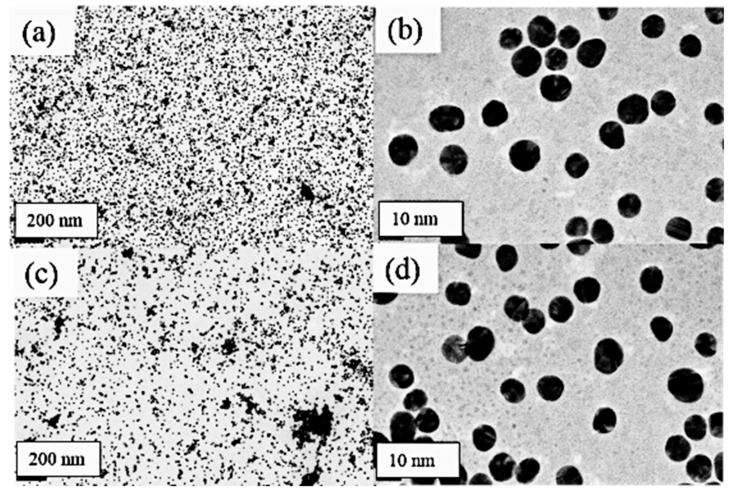
Transmission electron microscopy images of gold nanoparticles. (**a**) Control-200 nm, (**b**) control-10 nm, (**c**) 47 mM APTES-200 nm, and (**d**) 47 mM APTES-10 nm.

**Figure 9 nanomaterials-09-01417-f009:**
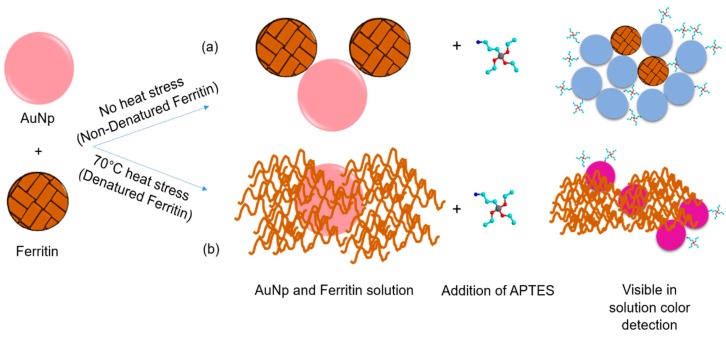
Displays the proposed mechanism for the blue shift observed in the ultraviolet visible absorbance spectra of the ferritin, gold nanoparticle, and APTES solutions as the ferritin is denatured.

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
