# Peer review of "Optical Detection of Denatured Ferritin Protein via Plasmonic Gold Nanoparticles Exposure through Aminosilane Solution"

_nanomaterials, 2019, doi:10.3390/nano9101417_

Round 1

Reviewer 1 Report

This paper shows that exposure of gold nanoparticles to denatured ferritin reduces the ability of APTES to promote their aggregation and hence to reduce the blue shift that accompanies aggregation.

The sensitivity of the peak absorbance to APTES vs ferritin concentration (native and denatured) should be presented as a graph in addition to the shown spectra

Please explain the physical significance of 'near' vs 'far' CD spectra especially the 'far' spectra

Please make it more clear in the figure captions whether the data refer to native or denatured ferritin

Author Response

Reviewer 1:

We thank the referee for useful comments and recommendation.

We have modified the manuscript according to the comments.

The sensitivity of the peak absorbance to APTES vs ferritin concentration...We have added an inset in Fig. 1 for comparison as suggested by the referee. Please explain the physical significance of 'near' vs 'far' CD spectra ---We have explained the meaning (which is of course well-know) and incorporated in the text. Figure caption is made clear as asked for denatured and non-denatured ferritin samples. We hope the present manuscript is acceptable for Nanomaterials.

Reviewer 2 Report

This work on the visual detection of denatured proteins via the use of gold nanoparticles and aminosilane is interesting and several analytical parameters have been examined. However, some points should be further considered. Please find specific comments here below:

-section 3.1: this section which is explaining in details the proposed mechanism of interaction among ferritin, gold NPs and APTES, should be moved at the beginning of the results section. This will improve the clarity of the work;

-Abstract: the sentence “Hence, we have demonstrated here by creating a platform for visual biosensing of denatured proteins that is fast, low cost, sensitive and user friendly by exploiting the plasmonic noble metal nanoparticles.” Needs a language correction, since it is not clear in this form;

-Abstract: the sentence “The degradation of ferritin degradation….” Needs correction;

-Introduction, last paragraph: the expression “will provide peace of mind…” is not adequate for a scientific publication;

-The authors should clarify with some examples if the obtained results would be reproducible with other proteins rather than ferritin; is the protein influencing the detection?

-Is the temperature a key parameter to be controlled during the visual detection (not for heat stressing the protein)?

-if the method is developed to detect the presence of denatured proteins within a therapeutic drug product, it would be interesting to understand if the visual detection is influenced by the other components of this therapeutic drug product.

Author Response

Reviewer 2:

We thank the referee for useful comments and recommendation.

We have modified the manuscript according to the comments.

We kept the details the proposed mechanism of interaction among ferritin, gold NPs and APTES are kept in Section 3.1 as it explains all the data all over the manuscript.

Abstract: The sentences in Abstract are modified as suggested by the referee.

Introduction, last paragraph: The wording is modified as suggested.

Examples if the obtained results would be reproducible with other proteins: We have included a sentence at the end of the Results and discussion giving some examples (with references).

Is the temperature a key parameter Yes, hence everywhere the temperature is mentioned. However, experiments are done at 20C (mentioned in the text).

If the method is developed.......: This is included at the end of the Conclusion.

We hope the current manuscript is suitable for publication in nanomaterials. 

Round 2

Reviewer 2 Report

The authors have considered most of the reviewer comments. However, some of the comments have been addressed in an approximate way.

To reply to the comment "if the method is developed to detect the presence of denatured proteins within a therapeutic drug product, it would be interesting to understand if the visual detection is influenced by the other components of this therapeutic drug product." the authors have added the reviewer sentence at the end of the manuscript. This is not acceptable: the response to the comment should be either results of some measurements or at least a discussion based on literature data.

For replying to the comment "

The authors should clarify with some examples if the obtained results would be reproducible with other proteins rather than ferritin; is the protein influencing the detection?" the authors have added some references, without a discussion on this topic. A discussion on this is necessary.

In view of these specific references added, it is necessary also to stress which are the differences and the advancement of the proposed manuscript with respect to the other ones which look very similar to this, apart from the protein to which the method is applied.

Author Response

Referee 2:

Thank you for the comments.

We have modified the text and added the necessary text regarding the two comments.

The modified texts are presented in Pages 13 and 14.

We hope the manuscript is acceptable for Nanomaterials.

Thank you.